# Hybrid Quantum-Classical Multi-Agent Pathfinding

**Thore Gerlach** [1 2]  **Loong Kuan Lee** [2]  **Frederic Barbaresco** [3]  **Nico Piatkowski** [2]

## Abstract

Multi-Agent Path Finding (MAPF) focuses on de-termining conflict-free paths for multiple agents navigating through a shared space to reach speci-fied goal locations. This problem becomes compu-tationally challenging, particularly when handling large numbers of agents, as frequently encoun-tered in practical applications like coordinating autonomous vehicles. Quantum Computing (QC) is a promising candidate in overcoming such lim-its. However, current quantum hardware is still in its infancy and thus limited in terms of comput-ing power and error robustness. In this work, we present the first optimal hybrid quantum-classical MAPF algorithms which are based on branch-and-cut-and-price. QC is integrated by iteratively solv-ing QUBO problems, based on conflict graphs. Experiments on actual quantum hardware and results on benchmark data suggest that our ap-proach dominates previous QUBO formulations and state-of-the-art MAPF solvers.

## 1. Introduction

Emerging domains of large-scale resource allocation prob-lems, such as assigning road capacity to vehicles, ware-house management or 3D airspace to *Unmanned Aerial Vehicles* (UAVs), often require *Multi-Agent Pathfind-ing* (MAPF) (Stern et al.; Choudhury et al.; Li et al., c) to determine feasible allocations. MAPF involves calculating non-colliding paths for a large number of agents simulta-neously, presenting significant computational challenges in realistically sized scenarios. These challenges are becoming increasingly relevant in case of large-scale real-world ap-plications. E.g., future UAV traffic in urban environments, driven by parcel delivery demands, is expected to involve managing thousands of flight paths (Doole et al., 2020).

---

[1]University of Bonn [2]Fraunhofer IAIS [3]Thales Land and Air Systems. Correspondence to: Thore Gerlach <gerlach@iai.uni-bonn.de>.

*Proceedings of the 42$^{nd}$ International Conference on Machine Learning*, Vancouver, Canada. PMLR 267, 2025. Copyright 2025 by the author(s).

The scalability of optimal state-of-the-art MAPF solvers is limited, since finding optimal solutions is NP-hard (Sharon et al.). Thus one often falls back to suboptimal or anytime methods (Li et al., a;b; Huang et al.; Okumura, 2023b;a; Li et al., 2021). Even though such methods are compu-tationally efficient in finding a feasible solution, the solu-tion quality can be insufficient which implies the urge for optimal solution methods. By reformulating MAPF as a multi-commodity flow problem, it can be solved optimally via *Integer Linear Programming* (ILP). However, the re-duction uses an inefficient representation of the problem setting in terms of space complexity and is only effective on small instances. More popular techniques for optimal MAPF include *Conflict-based Search* (CBS) (Sharon et al.) and *Branch-and-Cut-and-Price* (BCP) (Lam et al., b). CBS is a two-level procedure, with splitting a search tree based on detected conflicts between agents and subsequent replan-ning. This tree is explored with best-first search until a collision-free node is found. BCP takes a different approach of considering a (possibly infeasible) solution which is then refined iteratively by successively adding paths and con-straints. Similarly to CBS, BCP is a two-level algorithm: on the low level it solves a series of single-agent pathfinding problems, while the high level uses ILP to assign feasible paths to agents. While low level single agent pathfinding can be performed efficiently, the high level problems of both CBS and BCP remain NP-hard.

*Quantum Computing* (QC) is considered promising for tackling certain NP-hard *Combinatorial Optimization* (CO) problems. This is due to its potential to leverage quantum phenomena to solve certain types of problems faster than classical methods. The quantum mechanical effect of *su-perposition* enables exploration of a vast solution space in parallel, which is particularly advantageous for finding the best solution from a large set of possibilities. During this exploration quantum *entanglement* allows the encoding of high-order relationships among problem variables, leading to an effective exploration of the solution space. In par-ticular, QC is suited for solving *Quadratic Unconstrained Binary Optimization* (QUBO) problems (Punnen, 2022)

$$\min_{\boldsymbol{z} \in \{0,1\}^n} \boldsymbol{z}^\top \boldsymbol{Q} \boldsymbol{z} \,, \qquad (1)$$

where $\boldsymbol{Q}$ is an $n \times n$ real matrix and $n$ is the number of qubits. Despite this simple problem structure, it is NP-

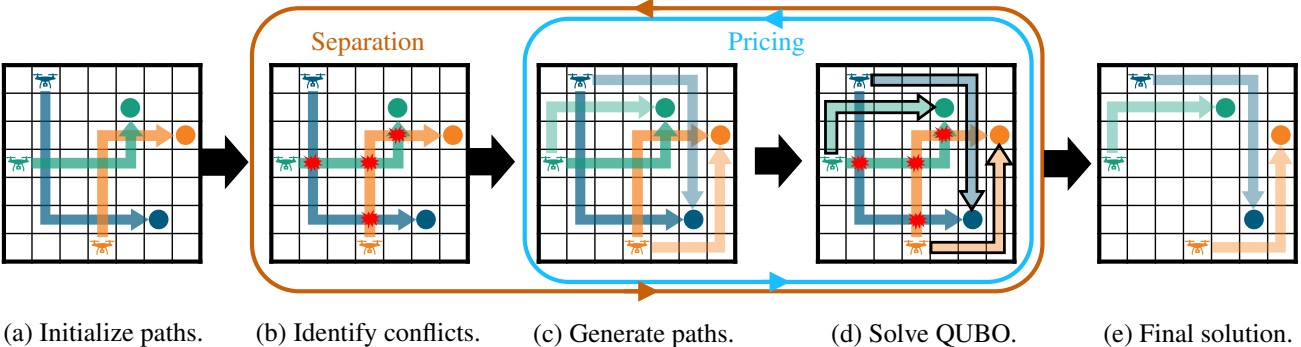

(a) Initialize paths.  (b) Identify conflicts.  (c) Generate paths.  (d) Solve QUBO.  (e) Final solution.

Figure 1: Schematic visualization of our quantum MAPF algorithm. (a) First, initial paths are generated for every agent with possible conflicts. (b) We enter the outer loop (separation), where we identify conflicts between paths and add them to the problem. (c) In the pricing step, we generate new paths for every agent and (d) find the best set of paths by solving a QUBO problem. This inner loop is repeated until adding a new path cannot improve the solution quality, while the outer loop is terminated when our set of chosen paths has no conflicts. (e) By construction, a conflict-free optimal set of paths is returned.

hard and hence encompasses a wide range of real-world problems, such as chip design (Gerlach et al., 2024), flight gate assignment (Stollenwerk et al., a) and trajectory planning (Stollenwerk et al., b).

One prominent QC method is *Adiabatic Quantum Computing* (AQC), which is grounded in the adiabatic theorem (Albash & Lidar, 2018). This theorem states that a quantum system will remain in its ground state if its Hamiltonian changes slowly enough and there is a sufficient energy gap from excited states. The problem is encoded into the final Hamiltonian, and the system is evolved gradually from an initial Hamiltonian. If done adiabatically, the system ends in the ground state of the problem Hamiltonian, which corresponds to the optimal solution. Approximations for AQC can be obtained by either using digital QC hardware, such as the *Quantum Approximate Optimization Algorithm* (Farhi et al., 2014), or by purposefully built analog devices, such as *Quantum Annealers* (Johnson et al., 2011). Despite QC's promise, we find ourselves in the *Noisy Intermediate-scale Quantum* (NISQ) (Preskill, 2018) era. QC devices face challenges including error rates, limited computing power and restricted hardware topology, which lead to suboptimal solutions obtained with currently available hardware.

In this paper, we investigate the use of hardware-aware QC for MAPF by constructing two optimal hybrid quantum-classical algorithms based on QUBO subproblems. To the best of our knowledge, these are the first quantum algorithms for MAPF. We call our algorithms *QUBO-and-Price* (QP) and *QUBO-and-Cut-and-Price* (QCP) which are based on the idea of BCP. We iteratively add paths (and constraints) to the problem and solve a QUBO, which leads to the applicability of QC. This gives us an upper bound on the best possible solution and a stopping condition tells us when our set of paths contains the optimal solution. An overview of QCP can be found in Fig. 1. Even though cur-

rent quantum hardware is still limited, our framework is modular and designed to be compatible with future devices.

Our contributions can be summarized as follows:

- Two novel optimal hybrid quantum-classical MAPF algorithms (QP and QCP), which iteratively solve restricted ILP problems by using QUBO,

- An optimality criterion with proof, generalizing the pricing problem of classical column generation,

- Hardware-aware QUBO formulations for the restricted ILP problems with using the concept of conflict graphs, leading to parallel solvable independent subproblems,

- Extensive experiments on benchmark datasets which show the superiority of our method over previous QUBO approaches on real quantum hardware and state-of-the-art MAPF solvers.

## 2. Related Work

Our algorithms are based on BCP (Lam et al., b). Despite their optimality, such algorithms require a sophisticated branching strategy and are not guaranteed to find a good solution in reasonable time. An anytime adaption of MAPF BCP has been investigated (Lam et al., a), but further investigation of optimal anytime algorithms is of great interest (Okumura, 2023a). Heuristics are used for avoiding exponentially many branching steps, leading to efficient suboptimal algorithms (Sadykov et al.). However, such rounding heuristics can lead to unsatisfying results, making the investigation of QC for this task intriguing.

QC is a promising candidate for large-scale planning problems (Stollenwerk et al., a; Li et al., 2024). In the area of multi-agent problems, flow-problem formulations have

been investigated (Ali et al., 2024; Zhang et al., 2021; Tarquini et al., 2024; Davies & Kalidindi). These methods are edge-based, that is each edge in the spatio-temporal graph is represented by a qubit. Even though certain constraints can be integrated into the quantum state in this framework, the problem size is way beyond current QC hardware capabilities and also infeasible for near-term devices.

Instead of representing all edges in the given graph, a different approach is taken in (Martín & Martin) by considering which path to choose as decision variables. This leads to introducing a QUBO formulation for the *Unsplittable Multi-Commodity Flow* problem by directly integrating the inequality constraints. Similar to BCP, they iteratively add paths to their problem. However, they present no theoretically sound criterion on when to stop, but have to rely on suboptimal heuristics. Furthermore, the large amount of constraints have to be incorporated into the QUBO formulation which either leads to a huge number of auxiliary variables or the need for an iterative optimization scheme to adapt Lagrangian parameters (Mücke & Gerlach, 2023). The authors of (Stollenwerk et al., b; Huang et al., 2022) circumvent this problem by using conflict graphs for representing possible constraints. However, their methods are not directly applicable to the MAPF setting, since only the starting times of preplanned trajectories are optimized.

We combine the ideas of an iteratively expanding path-based approach with the concept of conflict graphs. A pricing criterion tells us when all variables are included which are part of an optimal solution. The proof is based on (Rönnberg & Larsson), which however assumes negativity on reduced costs. We loosen this assumption and adapt it for general applicability to MAPF.

## 3. Background

For notational convenience, we onforth denote matrices as bold capital letters (e.g. $\boldsymbol{A}$), vectors as bold lowercase letters ($\boldsymbol{a}$) and define $\mathbb{B} := \{0, 1\}$. Furthermore, let $\boldsymbol{1}$ denote the vector consisting only of ones, with the dimension following from context. Lastly, let $\preceq$ denote entry-wise inequality between vectors, i.e., $\boldsymbol{a} \preceq \boldsymbol{b} \Leftrightarrow a_i \leq b_i \ \forall i$. The following sections will formalize MAPF (Sec. 3.1) and give a mathematical formulation of how to solve it (Sec. 3.2).

### 3.1. Multi-Agent Pathfinding

The input to the MAPF problem is a set of agents $A$, a weighted undirected graph $G = (V, E)$ and an origin-destination pair for each agent. The graph $G$ captures the underlying environment, where all possible agent states (e.g. location) are represented by $V$ and $E$ can be regarded as valid transitions from one state to another, with an underlying cost. $V$ already captures environmental constraints,

such as possible obstacles, while $E$ captures motion constraints of the agents, e.g., velocity and maximum turning rates of UAVs. The goal of MAPF is now to find optimal paths from the origin to the destination for each agent, s.t. they avoid pairwise conflicts. For rating optimality, we use the objective of the sum of all weighted path lengths. We here consider classical collision conflicts, that is the *vertex conflict* and the *(swapping) edge conflict*. A vertex conflict between two agents exists if they move to the same node at the same time, while an edge conflict prohibits two agents to use the same edge at the same time. Introducing the time component, we allow the agents to start at different points in time, while also giving them the opportunity to wait at a certain location.

For finding a solution to the MAPF problem, a spatio-temporal directed graph formulation is typically used. That is, we define $G_T = (V_T, E_T)$ as the graph, with nodes $v = (s, t) \in V \times \{1, \ldots, T\}$ and edges $e = (v, v') = ((s, t), (s', t+1)) \in V_T \times V_T$ with weights $w_e = w_{(s,s')}$, where $(s, s') \in E$ and $T \in \mathbb{N}$ is a maximum allowed time horizon. We define the reverse edge of $e = (s, t), (s', t+1)$ as $\bar{e} := (s', t), (s, t+1)$ for representing edge conflicts. $G_T$ is an acyclic weighted directed graph with $|V_T| = T|V|$ and $|E_T| = \mathcal{O}(ST|V|)$, where $S$ is the average number of possible state transitions. For example, if we consider the classical two-dimensional grid environment, an agent has five possible state transitions, i.e., wait, go north, go east, go south and go west.

Every agent $a \in A$ is obliged to find a path in $G_T$ from origin $(o_a, t_a) \in V_T$ to destination $(d_a, t_a + T_a) \in V_T$ for a starting time $t_a \in [T]$ and $T_a \leq T - t_a$. A path $p$ is a sequence of edges $p := (e_1, \ldots, e_{T_a-1})$, s.t. $e_t = ((x, t_a+t), (y, t_a+t+1))$, $e_{t+1} = ((y, t_a+t+1), (z, t_a+t+2))$, $e_1 = ((o_a, t_a), (s, t_a+1))$ and $e_{T_a} = ((s, t_a+T_a-1), (d_a, t_a+T_a))$. The cost/length $c_p$ of a path $p$ is defined as the sum of all its edge weights, i.e., $c_p := \sum_{e \in p} w_e$. Specifically, we go over to the mathematical description of the MAPF objective.

### 3.2. Path-Based Formulation

We encode each possible path $p \in \mathcal{P}$ for every agent by $z_p \in \mathbb{B}$. An ILP formulation is given by:

$$\text{MP}: \min_{\boldsymbol{z} \in \mathbb{B}^N} \sum_{a \in A} \sum_{p \in \mathcal{P}_a} c_p z_p \tag{2a}$$

$$\text{s.t.} \sum_{p \in \mathcal{P}_a} z_p = 1, \ \forall a \in A \tag{2b}$$

$$\sum_{a \in A} \sum_{p \in \mathcal{P}_a} x_v^p z_p \leq 1, \ \forall i \in V_T \tag{2c}$$

$$\sum_{a \in A} \sum_{p \in \mathcal{P}_a} (y_e^p + y_{\bar{e}}^p) z_p \leq 1, \ \forall e \in E_T, \tag{2d}$$

where $x_v^p, y_e^p \in \mathbb{B}$ indicate whether path $p$ visits vertex $v \in V_T$ / edge $e \in E_T$, $\mathcal{P}_a$ indicates the set of all possible paths for agent $a$ and $N = |\mathcal{P}|$, $\mathcal{P} = \bigcup_{a \in A} |\mathcal{P}_a|$. Note that $x_v^p, y_e^p$ are constant and we just optimize over $z_p$. The constraint in (2b) ensures that exactly one path is chosen for every agent, while (2c) and (2d) avoid conflicts. We denote (2) as the *Master Problem* (MP).

We have $|V_T|$ constraints for avoiding vertex conflicts and $|E_T|$ constraints for avoiding swapping conflicts. The number of our decision variables representing possible solutions now corresponds to the number of all possible paths for all agents, which is exponential in the maximum time horizon, $N = \mathcal{O}(|A|S^T)$. A large maximum time $T$ or number of agents $|A|$ can make the problem size increase very quickly, making it infeasible to solve it with current NISQ devices.

## 4. Methodology

Not only the huge number of decision variables poses a challenge for current quantum computers, but also the number of constraints strongly impacts the solvability of the resulting QUBO formulation. We overcome this issue by considering the *Restricted Master Problem* (RMP) which optimizes over a subset of decision variables

$$\text{RMP}: \min_{\boldsymbol{z} \in Z} \boldsymbol{c}^\top \boldsymbol{z} \tag{3a}$$

$$\text{s.t. } \boldsymbol{D}\boldsymbol{z} \preceq \mathbf{1}, \tag{3b}$$

with $Z := \{\boldsymbol{z} \in \mathbb{B}^n : \sum_{p \in P_a} z_p = 1, \ \forall a \in A\}$, $P_a \subset \mathcal{P}_a$, $n = |P|$, $P = \bigcup_{a \in A} P_a$ and $\boldsymbol{c} \in \mathbb{R}^n$ is the vector consisting of the corresponding path lengths. The constraint matrix $\boldsymbol{D}$ captures inequality constraints from (2):

$$D_{i,p} = \begin{cases} x_i^p, & \text{if } i \in V_T, \\ y_i^p + y_i^p, & \text{if } i \in E_T. \end{cases}$$

To get equivalence between (3) and (2), we use a two-loop iterative optimization scheme: the outer loop decides which constraints are not fulfilled and should be added to our problem and the inner loop optimizes over subsets of all possible paths for every agent, increasing the number of paths in every step (see Fig. 1 and Algorithm 1). This procedure builds with the hope that we already find a (nearly) optimal solution with not exploring the whole search space, both in terms of decision variables $n$ and number of constraints $m$. The mathematical framework describing this technique is called *Column/Row Generation* (Lübbecke). The columns are identified with the decision variables and the rows with the constraints. It alternatingly solves a high-level (RMP) and low-level *Pricing* and *Separation* problems (PP and SP), which decide which variables/constraints to add to the MP. If solving the PP/SP tells us to not add any more variables, the solution to the MP is optimal by construction.

---

**Algorithm 1** QUBOANDPRICEANDCUT

**Input:** Shortest independent paths $P$ and $\boldsymbol{z} = \mathbf{1}$
**Output:** Optimal feasible solution
1: $C \leftarrow \emptyset$
2: **while** $\boldsymbol{z}$ is infeasible **do**          ▷ Separation
3:     Add violated constraints to $C$
4:     **while** not (5) **do**          ▷ Pricing
5:         $p^* \leftarrow \arg\min_p \bar{c}_p(\boldsymbol{\lambda}, C)$   ▷ Shortest path
6:         $P \leftarrow P \cup \{p^*\}$
7:         $\boldsymbol{\lambda} \leftarrow \text{OPTIMIZELAGRANGIAN}(P, C)$  ▷ Solve (8)
8:         $\boldsymbol{z} \leftarrow \text{OPTIMIZEMASTER}(P, C)$  ▷ QUBO (Sec. 4.3)
9:     **end while**
10: **end while**

---

This method was developed for *Linear Programming* (LP) (Dantzig) without the restriction of integrality of the variables—in our case we do not optimize over the continuous domain $[0, 1]$ but over the binary values $\{0, 1\}$. To cope with these kind of ILP problems, *branching* methods, such as BCP (Desrosiers & Lübbecke), are used. In the worst case, exponentially many branching steps are needed. We circumvent this problem by considering solutions to the ILP MP, instead of the relaxed LP version. Ideally, we want $n \ll N$, while an optimum of RMP is also an optimum of MP. The next section governs how to add new paths to our problem and check whether a solution is equivalent.

### 4.1. Pricing

Instead of using a relaxed LP formulation for the PP, we use *Lagrangian Relaxation* (LR) (Wolsey, 2020), which can be used for ILP problems of the form in (3). The *partial Lagrangian* is given by $\mathcal{L}(\boldsymbol{z}, \boldsymbol{\lambda}) := \boldsymbol{c}^\top \boldsymbol{z} + \boldsymbol{\lambda}^\top (\boldsymbol{D}\boldsymbol{z} - \mathbf{1})$, $\boldsymbol{\lambda} \in \mathbb{R}_+^m$ and the LR of the RMP is defined as

$$\text{LR}: \mathcal{L}(\boldsymbol{\lambda}) := \min_{\boldsymbol{z} \in Z} \mathcal{L}(\boldsymbol{z}, \boldsymbol{\lambda}) = \min_{\boldsymbol{z} \in Z} \bar{\boldsymbol{c}}(\boldsymbol{\lambda})^\top \boldsymbol{z} - \boldsymbol{\lambda}^\top \mathbf{1}, \tag{4}$$

where $\bar{\boldsymbol{c}}(\boldsymbol{\lambda}) := \boldsymbol{c} + \boldsymbol{\lambda}^\top \boldsymbol{D}$ is the *Lagrangian reduced cost* vector. Note that $\mathcal{L}(\boldsymbol{\lambda}) \leq v(\text{RMP})$, where $v(\cdot)$ indicates the optimal value of the optimization problem.

**Theorem 4.1.** *Let $\hat{v}$ be the objective value of a feasible solution $\bar{\boldsymbol{z}}$ to RMP ($\boldsymbol{D}\bar{\boldsymbol{z}} \preceq \mathbf{1}$, $\bar{\boldsymbol{z}} \in Z$ and $\hat{v} = \boldsymbol{c}^\top \bar{\boldsymbol{z}}$) and $\boldsymbol{\lambda} \in \mathbb{R}_+^m$. If $v(\text{RMP}) \neq v(\text{MP})$ then*

$$\exists a \in A: \min_{p \in \mathcal{P}_a \backslash P_a} \bar{c}_p(\boldsymbol{\lambda}) - \min_{p \in P_a} \bar{c}_p(\boldsymbol{\lambda}) < \hat{v} - \mathcal{L}(\boldsymbol{\lambda}). \tag{5}$$

*Proof.* We give a proof by showing that $v(\text{RMP}) = v(\text{MP})$ holds if (5) does not hold. The path variables not in RMP are implicitly assumed to take the value $0$. Assume now, that one variable not included in the RMP takes value $1$ for

agent $a \in A$, i.e., $\sum_{p \in \mathcal{P}_a \setminus P_a} z_p \geq 1$. It follows that

$$
\begin{aligned}
v(\text{MP}) &= \min_{\boldsymbol{z} \in \mathcal{Z}} \left\{ \sum_{p \in \mathcal{P}} c_p z_p : \boldsymbol{D}\boldsymbol{z} \preceq \boldsymbol{1}, \sum_{p \in \bar{P}_a} z_p \geq 1 \right\} \quad \text{(6a)} \\
&\geq \min_{\boldsymbol{z} \in \mathcal{Z}} \left\{ \mathcal{L}(\boldsymbol{z}, \boldsymbol{\lambda}) : \sum_{p \in \bar{P}_a} z_p \geq 1 \right\} \quad \text{(6b)} \\
&= \min_{\boldsymbol{z} \in \mathcal{Z}} \left\{ \sum_{p \in \mathcal{P}} \bar{c}_p(\boldsymbol{\lambda}) z_p : \sum_{p \in \bar{P}_a} z_p \geq 1 \right\} - C_{\boldsymbol{\lambda}} \quad \text{(6c)} \\
&= \min_{\boldsymbol{z} \in \mathcal{Z}} \sum_{p \in \mathcal{P} \setminus \mathcal{P}_a} \bar{c}_p(\boldsymbol{\lambda}) z_p + \min_{p \in \bar{P}_a} \bar{c}_p(\boldsymbol{\lambda}) - C_{\boldsymbol{\lambda}} \quad \text{(6d)} \\
&= \sum_{b \in A \setminus \{a\}} \min_{p \in P_b} \bar{c}_p(\boldsymbol{\lambda}) + \min_{p \in \bar{P}_a} \bar{c}_p(\boldsymbol{\lambda}) - C_{\boldsymbol{\lambda}} \quad \text{(6e)} \\
&= \min_{p \in \bar{P}_a} \bar{c}_p(\boldsymbol{\lambda}) - \min_{p \in P_a} \bar{c}_p(\boldsymbol{\lambda}) + \mathcal{L}(\boldsymbol{\lambda}) \quad \text{(6f)} \\
&\geq \hat{v} \geq v(\text{RMP}) , \quad \text{(6g)}
\end{aligned}
$$

with $\bar{P}_a = \mathcal{P}_a \setminus P_a$, $C_{\boldsymbol{\lambda}} = \boldsymbol{\lambda}^\top \boldsymbol{1}$ and $\mathcal{Z} = \{\boldsymbol{z} \in \mathbb{B}^N : \sum_{p \in \mathcal{P}_a} z_p = 1, \forall a \in A\}$. (6g) holds since we assume that (5) does not hold. We follow that no feasible solution to MP with $\sum_{p \in \mathcal{P}_a \setminus P_a} z_p \geq 1$ can be better than an optimal solution to RMP for each $a \in A$. As all paths not included in the RMP are in the set $\bigcup_{a \in A} \mathcal{P}_a \setminus P_a$ and $v(\text{MP}) \leq v(\text{RMP})$ is always true, we conclude $v(\text{MP}) = v(\text{RMP})$. Hence, if $v(\text{MP}) \neq v(\text{RMP})$ then (5) holds. $\square$

This theorem leads to an optimality criterion, telling us when the variables in the RMP contain the ones for an optimal solution to the MP. The pricing problem (PP) aims to find a path not included in the RMP with optimal reduced costs

$$
\text{PP}: \min_{p \in \mathcal{P}_a \setminus P_a} \bar{c}_p(\boldsymbol{\lambda}), \forall a \in A . \quad \text{(7)}
$$

Even though $\mathcal{P}_a \setminus P_a$ can be exponentially large, (7) can be solved efficiently. It boils down to a $k$ *shortest path problem* ($k \leq |P_a|$) on the graph $G_T$ with updated edge weights—$\lambda_e$ is added to the cost of $w_e$ and $w_{\bar{e}}$ for conflicted edges and $\lambda_v$ is added to all incoming edges to conflicted vertices in the current solution $\bar{z}$. This can be done efficiently, e.g. using *Yen's algorithm* (Yen), which dynamically updates the new shortest path using the already computed paths, obtained with A*.

Interestingly, (5) generalizes the stopping criterion in classical column generation. The RHS is an upper bound on the optimality gap between the RMP and LD, i.e., $v(\text{RMP}) - v(\text{LD}) \leq \hat{v} - \mathcal{L}(\boldsymbol{\lambda})$. If that gap is 0, we exactly recover the criterion given in (Lam et al., b) given by $\bar{c}_p - \alpha_a < 0$, where $\alpha_a = \min_{p \in P_a} \bar{c}_p(\boldsymbol{\lambda})$ is the dual variable corresponding to the convexity constraint of agent $a$ (2b). In typical column generation, the pricing is solved

with an optimal dual solution of the LP relaxation of the RMP (Lam et al., b). In our case, we solve the *Lagrangian dual* (LD)

$$
\text{LD}: \max_{\boldsymbol{\lambda} \in \mathbb{R}_+^m} \mathcal{L}(\boldsymbol{\lambda}) = \max_{\boldsymbol{\lambda} \in \mathbb{R}_+^m} \min_{\boldsymbol{z} \in Z} \mathcal{L}(\boldsymbol{z}, \boldsymbol{\lambda}) . \quad \text{(8)}
$$

Due to standard ILP theory (Geoffrion, 2009), the optimum of LD and the LP relaxation of RMP coincides, since $Z$ is convex. Also, the dual parameters of this LP relaxation exactly correspond to the optimal $\boldsymbol{\lambda}^*$. Instead of relying on optimal Lagrange parameters, our theorem allows any choice of $\boldsymbol{\lambda} \in \mathbb{R}_+^m$.

### 4.2. Separation

In addition to adding new paths to our RMP, we can take a similar approach with handling the constraints. Starting off with no inequality constraints ((2c) and (2d)), we can iteratively add them to the RMP, reducing computational overhead. Given a feasible solution $\bar{z}$ to the RMP, we check whether it is also feasible for the MP. That is, we add the row/constraint $\boldsymbol{D}_i \bar{z} \leq 1$ to the RMP if $\boldsymbol{D}_i \bar{z} > 1$, $\forall i \in V_T \cup E_T$. However, to ensure that our algorithm is optimal, we need to find an optimal solution w.r.t. the current constraints. If this cannot be guaranteed, Algorithm 1 can diverge to a suboptimal solution. The performance of this procedure is evaluated in Sec. 5.

### 4.3. Solving RMP with QUBO

It remains to clarify how to obtain a solution of (3). In fact, solving the RMP (Algorithm 1, Line 8) is the main computational bottleneck of our developed algorithm, since finding shortest paths (Algorithm 1, Line 5) and solving an LP problem (Algorithm 1, Line 7) can be done efficiently. Since we want the solutions to be solvable with QC, we reformulate the constrained problems into QUBO formulations. We have a look at three different approaches, which all rely on using penalty factors to integrate constraints. The one-hot constraint given in $Z$ (2b) can be integrated by using

$$
\min_{\boldsymbol{z} \in Z} \boldsymbol{c}^\top \boldsymbol{z} \Leftrightarrow \min_{\boldsymbol{z} \in \mathbb{B}^n} \boldsymbol{c}^\top \boldsymbol{z} + \sum_{a \in A} \omega_o^a \left\| \boldsymbol{1}^\top \boldsymbol{z} - 1 \right\|^2 ,
$$

for large enough $\omega_o^a > 0$, which leads to a QUBO formulation (1). Setting $\omega_o^a > \max_{p \in P_a} c_p$ always guarantees the equivalence. It remains to incorporate the inequality constraints.

**Slack Variables** The first approach inserts slack variables for every inequality constraint, similar to (Davies & Kalidindi). This is often pursued in practice but can lead to large number of variables, especially when the number of constraints is large. The linear inequality constraint in (3) can be reformulated with using an auxiliary vector $\boldsymbol{s} \in \mathbb{B}^n$

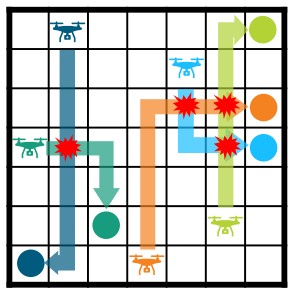
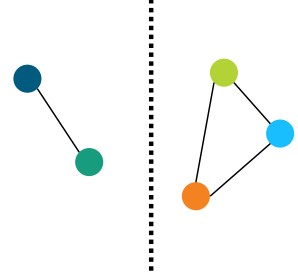

(a) Conflicted paths of exemplary MAPF problem.

(b) Corresponding disconnected conflict graph.

Figure 2: Schematic visualization of a conflict graph, which has two connected components. This leads to the decomposition into independent subproblems with reduced problem size.

with binary entries, since $\boldsymbol{Dz} \preceq \boldsymbol{1} \Leftrightarrow \boldsymbol{Dz} - \boldsymbol{1} + \boldsymbol{s} = \boldsymbol{0}$. This leads to the slightly different but equivalent problem $\min_{\boldsymbol{z} \in Z} \boldsymbol{c}^\top \boldsymbol{z}$, s.t. $\boldsymbol{Dz} - \boldsymbol{1} + \boldsymbol{s} = \boldsymbol{0}$. This constrained problem can then be reformulated to an equivalent unconstrained problem by introducing a penalty parameter $\omega_s > 0$

$$\min_{\boldsymbol{z} \in Z, \boldsymbol{s} \in \mathbb{B}^m} \boldsymbol{c}^\top \boldsymbol{z} + \omega_s \left\| \boldsymbol{Dz} - \boldsymbol{1} + \boldsymbol{s} \right\|^2 . \quad (9)$$

Using this formulation, however, comes with the overhead of introducing $m$ additional binary variables. One variable is needed for every inequality constraint, leading to a total QUBO dimension of $n + m$. As we are still in the NISQ era, it is better to resort to a QUBO formulation that uses fewer qubits. However, we have a look at the performance of real quantum hardware for this QUBO formulation in Sec. 5.

**Without Slack Variables** Since $\boldsymbol{D} \in \mathbb{B}^{m \times n}$, we can avoid using slack variables by using the equivalence

$$\min_{\boldsymbol{z} \in Z, \boldsymbol{s} \in \mathbb{B}^m} \left\| \boldsymbol{Dz} - \boldsymbol{1} + \boldsymbol{s} \right\| \Leftrightarrow \min_{\boldsymbol{z} \in Z} \left\| \boldsymbol{Dz} - \frac{1}{2}\boldsymbol{1} \right\| , \quad (10)$$

since $\min_{n \in \mathbb{N}, s \in \mathbb{B}} (n - 1 + s)^2 = \min_{n \in \mathbb{N}} (n - \frac{1}{2})^2 - \frac{1}{4}$. Thus, we do not optimize over any slack variables and reduce the corresponding QUBO size.

**Conflict Graph** As a third concept, we have a look into *Conflict Graphs* (CG). The vertices of such a graph correspond to all paths considered in the problem and the edges indicate whether there is at least one conflict between a pair of paths. An examplary schematic visualization of a conflict graph is given in Fig. 2. We denote the adjacency matrix of CG graph as $\boldsymbol{C}$. The RMP in (3) is equivalent to

$$\min_{\boldsymbol{z} \in Z} \boldsymbol{c}^\top \boldsymbol{z} + \omega_c \boldsymbol{z}^\top \boldsymbol{Cz} \quad (11)$$

where $\omega_c \in \mathbb{R}_+$ penalizes potential conflicts. Note that it is square and the dimension is only dependent on the number of considered paths and not on the number of constraints.

Since the structure of the CG depends on the underlying problem, it may contain several connected components. Thus, the graph of the QUBO matrix can have multiple connected components, dependent on $\boldsymbol{C}$. These connected components can be seen as smaller instances, giving us the ability to solve them independently. This can lead to large reduction of the considered problem sizes, which is very vital for current QCs, due to the limited number of available qubits. Furthermore, a lower density and well-behaved problem structure can have a huge effect on current quantum hardware, as we will see in Sec. 5.

This leads to two different hybrid quantum-classical MAPF algorithms. QUBO-and-Price (QP) describes generating new paths and stopping when (5) is violated (inner loop in Fig. 1). Even though the QUBO solver might not be optimal, this procedure leads to generating all sufficient paths for obtaining an optimal solution. We denote the extension of iteratively adding constraints to our problem as QUBO-and-Cut-and-Price (QCP). It is only guaranteed to generate an optimal path set if the QUBO is solved to optimality in every step. However, we also investigate its suboptimal performance in the next section.

## 5. Experimental Evaluation

We compare our algorithms QP and QCP with four state-of-the-art MAPF solvers: BCP (Lam et al., b), EECBS (Li et al., 2021), LaCAM* (Okumura, 2023a) and LNS2 (Li et al., b). BCP is an optimal algorithm but has also been adapted to anytime, while EECBS is suboptimal. LaCAM* and LNS2 are anytime algorithms, which can quickly obtain a good feasible solution. For more details on these methods, we refer to the original papers. The code was taken from their respective public repositories[1234]. We kept all standard parameters and set a maximum time limit of $180 \, s$ for all solvers. The experiments were conducted on a single core of the type Intel(R) Xeon(R) Silver 4216 CPU @ 2.10GHz.

For QP and QCP we generate the initial paths with *Prioritized Path Planning* (PPP) (Ma et al.). That is, every agent is planned successively, with removing edges and nodes of previously planned paths. No clever priorization heuristic is followed, we randomly sample which agent to be chosen next. We use a maximum limit of 30 pricing steps in total and compute optimal Lagrangian parameters $\boldsymbol{\lambda}^*$ with LP in every iteration. We compare the optimal solution of the RMP obtained by an ILP Brach-and-Bound method with the solutions obtained by two different QUBO solvers. As a classical baseline, we use the *Simulated Annealing* (SA)

---

[1]https://github.com/ed-lam/bcp-mapf
[2]https://github.com/Jiaoyang-Li/EECBS
[3]https://github.com/Kei18/lacam3
[4]https://github.com/Jiaoyang-Li/MAPF-LNS2

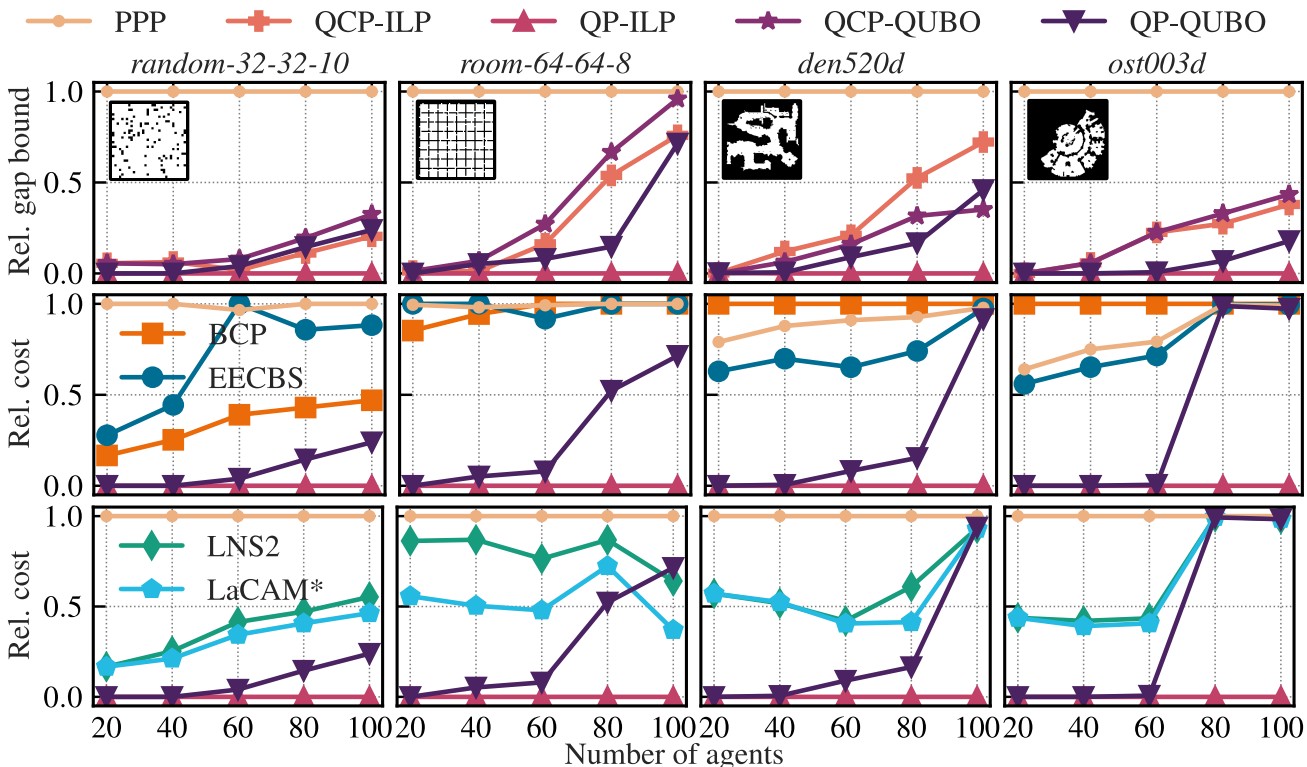

Figure 3: Relative performance comparison of our methods QP-ILP, QP-QUBO, QCP-ILP and QCP-QUBO to the baselines PPP, BCP, EECBS, LNS2 and LaCAM* on four different maps with a varying number of agents. The relative upper bound of the optimality gap (top) is shown along with the relative total path costs (middle and bottom) averaged over all 25 scenarios. The lower the better, i.e., a value of 0 corresponds to the best performance, while 1 corresponds to the worst performing algorithm.

implementation of D-Wave (D-Wave Systems Inc., 2025) and for real quantum hardware, we run experiments on a D-Wave Advantage_system5.4 (D-Wave Systems Inc., 2021) quantum annealer (QA). Since both solvers are probabilistic, we generate 1000 samples each and use 1000 Monte Carlo sweeps for SA and an annealing time of $20\,\mu s$ for QA. The remaining remaining unspecified parameters are set to their default values. We compare the three different QUBO formulations from Sec. 4.3 and denote them by SLACK (9), HALF (10) and CONFLICT (11). These QUBO formulations are dependent on penalty parameters, and understanding their impact is crucial for NISQ devices. Large penalty weights can increase the dynamic range of QUBO coefficients (Mücke et al.; Gerlach & Piatkowski, 2024) and decrease the spectral gap (Stollenwerk et al., a;b), reducing solution quality and requiring longer annealing times. In our experiments, we set these penalties to the values described in Sec. 4.3 for adhering the constraints and leave special tuning for future work.

As maps and instances, we use the well-known MovingAI benchmark (Stern et al.). This benchmark includes 33 maps and 25 random scenarios, some of which were utilized in our experiments. Every scenario on each map (with some

exceptions) consists of 1000 start-goal position pairs. To evaluate a solver on a given scenario, we run it on the first 20, 40, 60, 80 and 100 start-goal pairs for all 25 scenarios and indicate the average performance.

**Quantum Hardware Limitations**  We did not scale up our experiments further in terms of the number of agents (e.g. $|A| = 1000$) because the QUBO size grows rapidly with the number of paths and constraints, making the problem increasingly difficult to solve on current NISQ hardware. Even with conflict graph decomposition, the resulting QUBOs from larger environments often exceed the qubit capacity and connectivity limitations of existing quantum hardware.

While the D-Wave Advantage System features an impressive $5,670$ physical qubits[5], its limited qubit connectivity poses significant constraints. To embed arbitrary graphical structures, D-Wave relies on chaining multiple physical qubits into logical *super-qubits*, a process necessitated by the underlying hardware topology. The current Pegasus topology[6]

---

[5]https://www.dwavesys.com/solutions-and-products/systems
[6]https://www.dwavesys.com/media/jwwj5z3z/14-1026a-c_next-generation-topology-of-dw-quantum-processors.pdf

Table 1: Total path costs for different maps using 100 agents averaged over 25 scenarios. We show the mean and the standard deviation and indicate the best performing method in bold an the second best by underlining the result.

| Environment | LNS2 | LaCAM* | QP-QUBO | QP-ILP |
|---|---|---|---|---|
| *empty-32-32* | $2164.2 \pm 88.0$ | $\underline{2164.2 \pm 87.6}$ | $2166.5 \pm 88.9$ | $\mathbf{2163.4 \pm 88.6}$ |
| *random-32-32-10* | $2239.0 \pm 113.1$ | $2236.8 \pm 112.2$ | $\underline{2231.3 \pm 112.2}$ | $\mathbf{2225.4 \pm 111.5}$ |
| *room-64-64-8* | $6301.2 \pm 307.2$ | $\underline{6219.2 \pm 285.4}$ | $6323.4 \pm 342.4$ | $\mathbf{6107.1 \pm 303.7}$ |
| *maze-32-32-4* | $\underline{4725.4 \pm 334.6}$ | $\mathbf{4534.3 \pm 219.8}$ | $6332.5 \pm 1272.6$ | $5715.9 \pm 1020.6$ |
| *den312d* | $5499.4 \pm 232.6$ | $5492.3 \pm 224.9$ | $\underline{4559.0 \pm 195.2}$ | $\mathbf{4558.9 \pm 195.2}$ |
| *ost003d* | $15164.8 \pm 782.8$ | $\underline{15164.2 \pm 782.2}$ | $15166.2 \pm 783.0$ | $\mathbf{14901.8 \pm 1452.7}$ |
| *den520d* | $17391.3 \pm 970.8$ | $17391.0 \pm 970.8$ | $\underline{17389.5 \pm 973.5}$ | $\mathbf{17378.9 \pm 3541.7}$ |

supports embeddings of up to 182 densely connected logical qubits, while the upcoming Zephyr[7] topology extends this to 232. Combined with inherent noise limitations, this means that only a fraction of the full qubit count can be effectively utilized, restricting practical problem sizes to a few hundred logical qubits at most. Since path variables are added in every iteration of our algorithm, considering problems with more than 100 agents would thus lead to unreasonable results to the aforementioned limitations. As the quantum hardware matures, more large-scale experiments will be conducted in future work.

**Algorithm Performance**  In Fig. 3, we depict the performance of our two algorithms QP and QCP for four different maps of the MovingAI benchmark and varying the number of agents. All performances are shown relative to the best and worst performing configuration, i.e., $v_{\mathrm{rel}} := (v - v_{\mathrm{best}})/(v_{\mathrm{worst}} - v_{\mathrm{best}})$ and averaged over all 25 scenarios. This maps all obtained values to the range $[0, 1]$, where 0 indicates the best and 1 the worst performance, respectively. For QC-QUBO and QCP-QUBO we use the CONFLICT formulation and the SA solver.

The top plot row shows the mean relative upper bound $\hat{v}_{\mathrm{rel}} - v_{\mathrm{rel}}(\mathrm{LD})$ on the optimality gap for a different number of agents. It is not only a measure of solution quality, but it also quantifies the problem size. The higher this gap is, the more new paths are added to our problem during pricing, since it corresponds to the RHS in our optimality criterion (5). We can see that optimally solving the RMP (QCP-ILP and QP-ILP) often has a smaller gap than generating a possibly suboptimal solution by a QUBO solver. However, the QUBO methods largely improve upon the base PPP method. Even though QP clearly outperforms QCP for optimal solving (ILP) of RMP, QCP takes way less constraints into account and is thus computationally

more efficient. This is also beneficial for the QUBO solvers, since they can easier generate good solutions for a more well-behaved problem (less constraints).

The mean relative total path cost for the single MAPF instances are depicted in the middle and bottom plot row. We compare the baselines PPP, BCP, EECBS, LNS2 and LaCAM* with QP solving the RMP optimally (QP-ILP) and with QUBO (QP-QUBO). If a method does not return any solution in the given time window, we set its performance to the worst other performing method, allowing for a naive anytime comparison. It is evident that QP-ILP is always optimal, while QP-ILP almost always outperforms the best performing baselines LNS2 and LaCAM*, which are better for 100 agents on *room-64-64-8*. The (sub)optimal algorithms BCP and EECBS are always outperformed by our methods due to their bad anytime performance. It is interesting that for no map they are able to find all optimal solutions in the given time frame. Since QP-QUBO is able to outperform LNS2 and LaCAM* in many cases, our method is applicable without the need of exactly solving the RMP.

In Table 1, we depict the mean and standard deviation of the total path costs averaged over 25 scenarios for different maps using 100 agents. It is evident, that our method with solving the RMP exactly (QP-ILP) is almost always optimal, except for *maze-32-32-4*. We use PPP for initializing the set of paths which can lead to bad initial results for a large number of agents—especially for small corridors appearing in the maze-like maps. Adapting the initialization method for obtaining a valid set of conflict-free paths (e.g. using LNS2 or LaCAM* instead) can mitigate this effect. Further, finding valid solutions to the RMP with our proposed QUBO formulation (QP-QUBO) is often also outperforming state-of-the-art heuristics (LNS2 and LaCAM*). However, an increasing agent count makes it harder to solve the resulting high-dimensional QUBO problem, sometimes leading to unsatisfactory solver performance. With quantum hardware becoming more mature in the future, we expect the

---

[7]https://www.dwavesys.com/media/2uznec4s/14-1056a-a_zephyr_topology_of_d-wave_quantum_processors.pdf

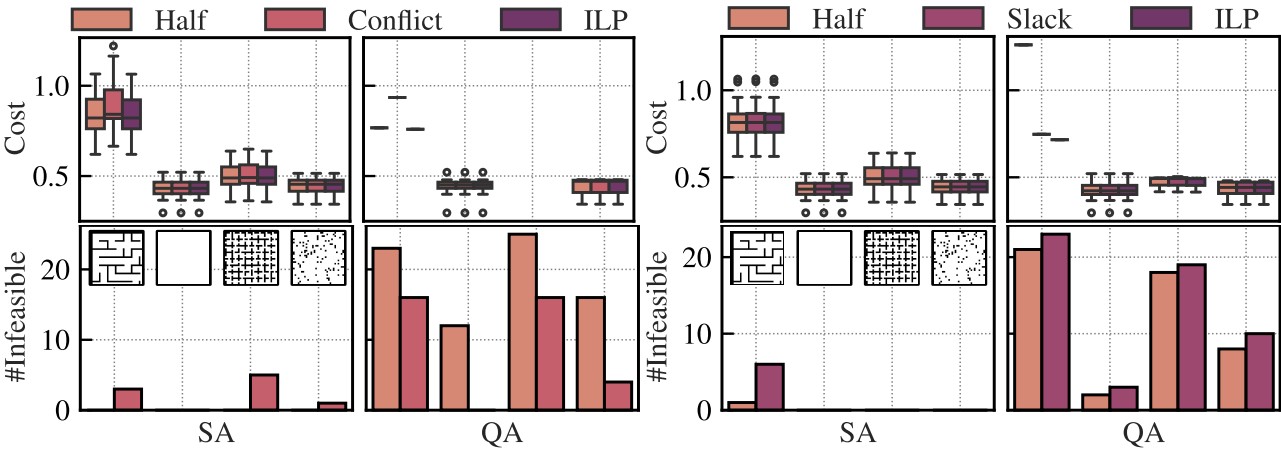

Figure 4: Performance comparison of different QUBO formulations for four different maps with 20-agent problems along with the optimal solution, where we run QP (Sec. 5) and QCP (Sec. 5) for 30 pricing steps. We compare SA and QA by indicating the total path cost of the best sample (top) and the number of infeasible solutions (bottom), i.e. (2c) and (2d) are violated. The cost is scaled by $10^{-3}$.

performance of QP-QUBO to approach the one of QP-ILP.

The most time consuming step in our algorithm is to solve the QUBO problem. However, a wall-clock time comparison is difficult for quantum devices nowadays, since there is a large communication overhead when using real quantum hardware over cloud services. Nevertheless, assuming perfect communication with the QA, we can generate solutions to a QUBO with an annealing time of around $50\,\mu s$. Due to its probabilistic nature, we have to generate only few thousand samples. In the near future, this could also lead to very good wall-clock time performance.

**QUBO Comparison** The effect of using different QUBO formulations for solving the RMP is depicted in Fig. 4. We consider four different maps (*maze-32-32-4*, *empty-32-32*, *random-32-32-10*, *room-32-32-4*) and compare the results of QP (Sec. 5) and QCP (Sec. 5) for 20 agents. This leads to varying QUBO sizes between 20 and 400, depending on the underlying map and scenario. We use SA and QA for solving HALF and CONFLICT for QP and HALF and SLACK for QCP.

In the top row, we show the cost of the best sample obtained by SA and QA over all 25 scenarios, along with the optimal solution (ILP). Only feasible samples are indicated here, that is only those who adhere the constraints. For QP, we can see that finding a feasible solution with the HALF QUBO is nearly almost optimal while the solution quality of CONFLICT slightly deteriorates. Comparing HALF and SLACK for QCP, we find that both solvers are able to find optimal solutions. However, QA has problems finding feasible solutions for the first map for all QUBO formulations.

The number of infeasible solutions returned by SA and QA

is depicted in the bottom row. While SA always finds feasible solutions for HALF, it is easier for QA to obtain feasibility with CONFLICT. This is due to the sparsity advantage of CONFLICT over HALF and the corresponding decomposition into independent subproblems. Since the hardware topology of current quantum computers is strongly limited, such properties have a large effect on the solution quality obtained. For QCP, we find that QA finds slightly more feasible solutions with HALF than SLACK. However, we note that only a few separation steps happened and thus only a small number of constraints have been generated. Using more pricing steps or scaling up the problem size would lead to way more included constraints, making the SLACK QUBO infeasible to solve.

## 6. Conclusion

In this paper, we presented two novel optimal hybrid quantum-classical MAPF algorithms. Extending the classical approach of Branch-and-Price-and-Cut, we circumvent exponentially many branching steps. Paths and constraints are iteratively added to our problem which is then solved by utilizing different QUBO formulations, making our algorithms suited for quantum computing. On an ideal adiabatic quantum processor, the QUBO subproblems are solved to global optimality. We proof a generalization of the classically known criterion telling us that our currently available path set is optimal. Experiments indicate good performance of our algorithm compared to state-of-the-art MAPF algorithms. Evaluating different QUBO formulations indicates the superiority of our approaches over previously presented methods. With these insights, we conclude that the hardware-aware design of the QUBO problems can already lead to an advantage on near-term quantum devices.

## Acknowledgments

This research has been funded by the Federal Ministry of Education and Research of Germany and the state of North-Rhine Westphalia as part of the Lamarr Institute for Machine Learning and Artificial Intelligence.

## Impact Statement

This paper presents work whose goal is to advance the fields of Optimization and Artificial Intelligence. There are many potential societal consequences of our work, none which we feel must be specifically highlighted here.

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
