# OpenReview forum: "Hybrid Quantum-Classical Multi-Agent Pathfinding"
_ICML.cc/2025/Conference — ICML 2025 poster_

### Official Review · Reviewer_32og · 2025-03-11

**Overall Recommendation:** 4

**Summary:**

This paper studies the problem of multi-agent pathfinding. The authors proposed a framework that can formulate the problem to a two-level optimization problem and then be able to use a quantum computing method to efficiently solve the problem (QUBO). The authors proved that the algorithm is able to find the optimal solution in the end. The authors provided a detailed comparison between the performance of the proposed methods and several benchmark planners' anytime performances and conclude that the proposed planner can efficiently solve MAPF problem optimally in many different cases.

**Claims And Evidence:**

As discussed below, there are other SOTA algorithms in the field, and it would be better if the authors can take them into consideration and compare the performance. Besides that, the claims are supported well and look good to me. Besides, the contribution of this work is not explicit to me. It seems that the core weapon of the framework - QUBO solving strategy is not the novelty here, and this work applies this method in the MAPF problem. The flow-problem formulations and integer programming related methods and formulations are studied before as well. It would be nice if the authors can explicitly discuss the contribution of this work more clearly.

**Essential References Not Discussed:**

As mentioned in the above Methods And Evaluation Criteria part, LaCAM series algorithms should be included and discussed. The papers are published in AAAI and IJCAI 2023, with the prior work PIBT published in IJCAI 2019.

**Experimental Designs Or Analyses:**

The authors used many experiments and illustrated the advantages of the proposed algorithms against BCP and LNS in different cases. The experiments look good in general, but in the performance comparison experiments, it would be interesting to see the results in experiments with more agents. As the authors mentioned on page 4, the efficiency of the quantum algorithm is closely related to the number of decision variables. Currently the maximum scale of the problem is 100 agents, and it would be nice if the authors can provide the results for experiments with more agents (1000, let's say) and compare the performance.

**Methods And Evaluation Criteria:**

In this paper, the authors compared the performance of the proposed planner with LNS2 and BCP. However, the current SOTA algorithm of MAPF problem should be LaCAM series algorithm. Specifically, LaCAM* is also an anytime MAPF algorithm which is able to converge to the optimal solution. Given this, it would be good if the authors can include LaCAM into the benchmarks and compare the performance with it (them). Besides, some variant algorithms of CBS are bounded suboptimal, such as EECBS (AAAI 2021), although it's not anytime, since BCP is taken into account, it would be nice if such bounded suboptimal algorithms can also be compared.

**Other Comments Or Suggestions:**

The term NISQ is used in page 4 without explanation. The explanation is on page 6. It's better to put the explanation before using it.

**Other Strengths And Weaknesses:**

The paper is written and organized clearly and the illustrations are useful for understanding the algorithm.

**Questions For Authors:**

All discussed in the above parts

**Relation To Broader Scientific Literature:**

In this paper, the authors proposed an interesting and applicable approach to solve MAPF problems using quantum computation, which is pretty novel in the community. On the other hand, the result in the paper paves the way for using quantum computing in more robotics scenarios, which is interesting and promising.

**Theoretical Claims:**

The proof looks good to me.

---

> ### Author Rebuttal · Authors · 2025-03-31
>
> We thank the reviewer for the constructive feedback and thoughtful observations. Below, we address each of the main concerns and suggestions raised.
>
> ### Clarification of Contributions and Novelty:
> We appreciate the reviewer’s request for a clearer articulation of our contributions. Our key novelty lies in proposing the first quantum-compatible algorithm for MAPF that is theoretically grounded and hardware-aware. While QUBO as a general optimization framework is not new, our contribution lies in:
> - Structuring the MAPF problem via a two-level optimization framework suitable for QUBO translation.
> - Proving a formal optimality guarantee in this setting, extending classical results to the integer domain.
> - Introducing a hardware-aware QUBO design, using conflict graphs to enable decomposition into independent subproblems for practical execution on current quantum devices.
>
> Quantum annealing is not yet outperforming classical methods in most practical MAPF scenarios. Our paper does not claim otherwise. Instead, our goal is to present a hardware-aware, theoretically grounded framework that integrates quantum optimization in a meaningful way—even if its benefits will become more tangible as hardware improves. We will clarify this positioning and explicitly reference hardware roadmaps such as IBM’s quantum roadmap (https://www.ibm.com/roadmaps/quantum/) to illustrate the forward-looking relevance of this work.
>
> We will revise the introduction and contribution sections to make these points more explicit.
>
> ### Comparison with SOTA MAPF Algorithms (LaCAM and EECBS):
> We thank the reviewer for pointing out important recent advancements. We are currently running experiments with LaCAM* and EECBS to compare against our proposed methods and will include the results or a detailed discussion in the updated version. Additionally, we will expand the related work section to cover the LaCAM series (including PIBT) and EECBS to provide better contextualization within the MAPF literature.
>
> ### Scaling Beyond 100 Agents (e.g., to 1000 Agents):
> We agree that scaling is a crucial topic. While our current experiments evaluate up to 100 agents, scaling to thousands of agents on today’s quantum hardware remains infeasible due to hardware limitations in terms of qubit count, connectivity, and precision. Nonetheless, our framework is modular and designed to be compatible with future quantum devices. We identify large-scale testing (e.g., 1000 agents) as an important direction for future work, especially as quantum hardware evolves.
>
> We appreciate the reviewer’s positive comments on the novelty and applicability of our approach, as well as the clarity and illustrations in the paper. We believe that with these improvements, the paper will provide even greater value to both the MAPF and quantum optimization communities.

---

> > ### Comment · Reviewer_32og · 2025-04-01
> >
> > I would like to sincerely thank the authors for their comprehensive rebuttal. I understand that under the current constraint of quantum hardware, it would be hard for the quantum-based MAPF algorithm to outperform the current methods, and the contribution of this work lies more in creating such a framework that could be useful in practice in the future when the hardware becomes better.
> >
> > I know that the authors are running the experiments, and it would be nice if the authors could provide the experimental data and demonstrate and compare the results to prove that the framework you proposed is useful in practice. Alternatively, in other ways, explicitly point out that the current hardware condition is the bottleneck of which part, and show how different hardware can greatly impact the performance and foresee what kind of equipment, which is reasonable to achieve in the future, can hopefully reach and outperform the current best classical methods based on a back-of-the-evenlop model or other techniques. The potential of this method could be supported by doing this kind of analysis.

---

> > > ### Author Response · Authors · 2025-04-04
> > >
> > > We thank the reviewer for recognizing the forward-looking nature of our framework and for this thoughtful suggestion. In response, we now include a two-part addition to the revised manuscript:
> > >
> > > ### Empirical Evaluation
> > >
> > > We ran experiments with EECBS, LaCAM and LaCAM* on four benchmark MovingAI maps discussed in the paper.
> > > Similar to the already evaluated BCP and LNS2, we used a maximum time limit of 180s and used default parameters otherwise.
> > > The same setup as in the paper is used, that is we evaluate each algorithm for 25 different scenarios for 20, 40, 60, 80 and 100 agents.
> > > We compare solving the RMP optimally with ILP (QP-ILP) and with solving the QUBO formulation with simulated annealing (QP-QUBO).
> > > Detailed results indicating mean and standard deviation over 25 different scenarios are given in the tables below.
> > >
> > > Considering the mean performance over all 25 scenarios, we observe that our method QP-ILP performs best in 17/20 cases.
> > > It is evident QP-QUBO performs only slightly worse, with also achieving best performance over all baselines in 15/20 cases.
> > > Near future quantum optimization would allow for a performance lying between optimally solving (QP-ILP) and simulated annealing (QP-QUBO).
> > > LaCAM* performs best in 3/20 cases which is probably due to a bad choice of initial paths for our algorithm.
> > > Due to our algorithms' anytime property, it can be combined with other algorithms to also find better initializations (e.g. with LaCAM*).
> > >
> > > *random-32-32-10*
> > >
> > > | #Agents | LNS2 | EECBS | LaCAM | LaCAM* | QP-ILP | QP-QUBO |
> > > | :-: | :-: | :-: | :-: | :-: | :-: | :-: |
> > > | 20 | 447.1 ± 40.9 | 448.0 ± 40.9 | 453.0 ± 41.0 | 447.1 ± 40.9 | **447.0 ± 40.8** | **447.0 ± 40.8** |
> > > | 40 | 887.3 ± 66.0 | 891.0 ± 65.9 | 910.5 ± 68.2 | 887.1 ± 66.0 | **886.3 ± 66.3** | **886.3 ± 66.3** |
> > > | 60 | 1332.9 ± 77.6 | 1344.9 ± 78.4 | 1388.7 ± 88.5 | 1332.4 ± 77.4 | **1329.6 ± 77.1** | 1330.0 ± 77.0 |
> > > | 80 | 1780.9 ± 95.6 | 1807.4 ± 95.0 | 1880.6 ± 96.3 | 1779.9 ± 94.3 | **1773.6 ± 93.2** | 1775.8 ± 93.9 |
> > > | 100 | 2239.0 ± 113.1 | 2288.5 ± 111.1 | 2410.2 ± 126.5 | 2236.8 ± 112.2 | **2225.4 ± 111.5** | 2231.3 ± 112.2 |
> > >
> > > *maze-32-32-4*
> > >
> > > | #Agents | LNS2 | EECBS | LaCAM | LaCAM* | QP-ILP | QP-QUBO |
> > > | :-: | :-: | :-: | :-: | :-: | :-: | :-: |
> > > | 20 | 862.8 ± 108.5 | 879.6 ± 112.0 | 883.4 ± 102.0 | 851.8 ± 105.8 | **835.4 ± 104.0** | **835.4 ± 104.0** |
> > > | 40 | 1778.0 ± 163.3 | 1824.2 ± 170.5 | 1828.3 ± 533.8 | 1753.8 ± 163.4 | **1724.1 ± 195.7** | 1743.8 ± 235.2 |
> > > | 60 | 2691.1 ± 264.7 | 2791.2 ± 219.4 | 2794.7 ± 1066.7 | **2623.0 ± 191.1** | 2626.3 ± 591.4 | 2696.0 ± 459.2 |
> > > | 80 | 3664.6 ± 226.3 | 3869.7 ± 195.4 | 3889.0 ± 1309.4 | **3575.6 ± 196.9** | 3778.4 ± 495.4 | 4234.9 ± 745.5 |
> > > | 100 | 4725.4 ± 334.6 | 5059.3 ± 240.4 | 5086.5 ± 2551.3 | **4534.3 ± 219.8** | 5715.9 ± 1020.6 | 6332.5 ± 1272.6 |
> > >
> > > *room-64-64-8*
> > >
> > > | #Agents | LNS2 | EECBS | LaCAM | LaCAM* | QP-ILP | QP-QUBO |
> > > | :-: | :-: | :-: | :-: | :-: | :-: | :-: |
> > > | 20 | 1208.0 ± 125.9 | 1216.3 ± 127.0 | 1226.6 ± 128.2 | 1203.3 ± 125.2 | **1194.7 ± 124.9** | **1194.7 ± 124.9** |
> > > | 40 | 2455.2 ± 187.5 | 2504.8 ± 189.3 | 2525.9 ± 188.1 | 2434.2 ± 180.0 | **2405.2 ± 178.2** | 2408.2 ± 179.3 |
> > > | 60 | 3673.8 ± 239.5 | 3794.4 ± 258.0 | 3847.4 ± 256.8 | 3634.8 ± 222.6 | **3569.2 ± 216.6** | 3580.2 ± 218.7 |
> > > | 80 | 4992.8 ± 259.1 | 5215.0 ± 307.7 | 5292.2 ± 287.5 | 4927.2 ± 252.3 | **4790.0 ± 969.0** | 4835.7 ± 264.0 |
> > > | 100 | 6301.2 ± 307.2 | 6644.0 ± 342.3 | 6794.6 ± 321.1 | 6219.2 ± 285.4 | **6107.1 ± 303.7** | 6323.4 ± 342.4 |
> > >
> > > *den312d*
> > >
> > > | #Agents | LNS2 | EECBS | LaCAM | LaCAM* | QP-ILP | QP-QUBO |
> > > | :-: | :-: | :-: | :-: | :-: | :-: | :-: |
> > > | 20 | 1059.2 ± 114.0 | 1062.9 ± 115.1 | 1082.0 ± 121.3 | 1059.4 ± 114.1 | **900.5 ± 117.3** | **900.5 ± 117.3** |
> > > | 40 | 2169.0 ± 141.9 | 2183.3 ± 143.3 | 2249.4 ± 160.9 | 2168.8 ± 142.0 | **1815.0 ± 127.5** | **1815.0 ± 127.5** |
> > > | 60 | 3247.2 ± 184.7 | 3287.6 ± 190.8 | 3414.6 ± 208.3 | 3244.6 ± 184.8 | **2727.0 ± 173.7** | **2727.0 ± 173.7** |
> > > | 80 | 4369.3 ± 203.1 | 4444.8 ± 211.3 | 4645.2 ± 252.5 | 4364.2 ± 203.1 | **3627.8 ± 161.2** | **3627.8 ± 161.2** |
> > > | 100 | 5499.4 ± 232.6 | 5641.4 ± 241.1 | 5902.4 ± 273.9 | 5492.3 ± 224.9 | **4558.9 ± 195.2** | 4559.0 ± 195.2 |
> > >
> > > ### Hardware Bottleneck and Forecasting Analysis
> > >
> > > To supplement the above empirical results, we provide a detailed analysis of quantum hardware constraints hindering
> > > evaluating scenarios with thousands of agents and discuss future projections.
> > > - Identification of the main algorithmic bottleneck: Finding a valid solution to the RMP
> > > - Identification of the main hardware bottlenecks: QUBO embedding limitations (qubit count, connectivity) and sample quality (limited by noise and precision).
> > > - A scaling analysis of our problem formulation in terms of qubit requirements (showing how many qubits are needed).
> > > - Reference to realistic near-term hardware improvements, e.g. IBM’s roadmap, and estimates of what scale of MAPF problem such devices might support

---

### Official Review · Reviewer_f7fA · 2025-03-13

**Overall Recommendation:** 3

**Summary:**

The paper proposes novel hybrid quantum-classical algorithms leveraging quantum annealers for the problem of multi-agent path finding (MAPF). It proposes two iterative variant algorithms, QUBO-and-Price (QP) and QUBO-and-Cut-and-Prince (QCP) based on the idea of branch-and-cut-and-price (BCP) to find conflict-free paths for multiple agents navigating through shared space, which are among the first of its kind in the literature. Alongside theoretical proof guarantee, the experiments show that these algorithms have achieved competitive performance comparatively. This work provides an interesting method for solving this problem.

**Claims And Evidence:**

below expectations

**Essential References Not Discussed:**

n/a

**Experimental Designs Or Analyses:**

below expectations

**Methods And Evaluation Criteria:**

n/a

**Other Comments Or Suggestions:**

n/a

**Other Strengths And Weaknesses:**

Strength:
1) The first quantum-classical solver for MAPF;
2) Detailed math formulation of the MAPF problem as a QUBO which can run on annealers;
3) Extensive experimental evaluations of the protocol which showed great performance compared to other solvers adopted for benchmark.

Weaknesses:
1) The proper tries to address the problem of coordinating paths for multiple agents. My question is then how is this related to machine learning as ICML is a conference for machine learning? I think it could be better presented at a conference like IROS;
2) The writing is not easy and logical enough to follow and requires special attention;
3) Some statements on the background of quantum computing are inaccurate. For example, in L84 - 88, the adiabatic theorem only guarantees the ground state if there are no thermo-perturbations which does not exist for D-Wave annealers. This statement could be misleading;
4) in L194, the "NISQ" device normally refers to gate-based methods, but not for annealers. Specifically, annealers do not suffer from the noise that is commonly referred to in gate-based models;
5) Some details require further explanation. For example, from (9) to (10), why does (10) hold if slack variables are not used? I have to think and guess it makes sense, but it can benefit from a more thorough explanation;
6) In the formulation of the QUBO problem, the quadratic nature comes from the way that the constraint is incorporated, not the original objective function. So there raises a question: How justifiable is it to use the annealer in this case. In other words, if the constraint is not enforced via a soft constraint but through some other form, the objective is just a linear programming problem, in which there exist many more specific solvers. A more detailed comparison with these solvers is lacking in the experiment;
7) Some ablation studies are missing. How does the penalty impact the problem accuracy and the hardware-specific sparsity? Probably also the chain broke during the embedding.

**Questions For Authors:**

n/a

**Relation To Broader Scientific Literature:**

n/a

**Theoretical Claims:**

inaccurate

---

> ### Author Rebuttal · Authors · 2025-03-31
>
> We thank the reviewer for the thoughtful and constructive feedback. Below, we address each of the raised concerns and clarify several technical points. We will incorporate any remaining clarifications and adjustments into the final version.
>
> ### Relevance to ICML and the Machine Learning Community:
> We acknowledge the reviewer's concern regarding scope. While MAPF is a combinatorial optimization problem, our submission aligns with ICML’s growing interest in quantum computation, optimization theory, and AI planning. Recent ICML papers have explored quantum optimization, hardware-aware algorithm design, and combinatorial planning—topics closely related to our work. Furthermore, MAPF is widely studied in multi-agent reinforcement learning and planning under constraints, often intersecting with machine learning methodologies.
>
> ### Clarification on the Adiabatic Theorem (Lines 84–88):
> We agree that the current phrasing may overstate the practical applicability of the adiabatic theorem to real-world quantum annealers. We will revise this section to clearly distinguish idealized adiabatic quantum computation from practical implementations such as D-Wave’s quantum annealer, which is subject to thermal noise and device-specific imperfections. Our discussion will be reframed under the broader umbrella of quantum optimization, and not restricted to quantum annealing.
>
> ### Use of the Term “NISQ” (Line 194):
> We appreciate the clarification. While “NISQ” is typically used to describe gate-based devices, it is also used in broader literature to refer to pre-fault-tolerant quantum systems, including annealers. We will clarify this distinction and highlight that quantum annealers suffer from different noise models (e.g., Integrated Control Errors, ), as documented in D-Wave’s technical materials (https://docs.dwavequantum.com/en/latest/quantum_research/errors.html). Regardless, we will revise this terminology for clarity and consistency.
>
> ### Use of QUBO for ILP and Comparison to Classical Solvers:
> We acknowledge the need to clarify this point. The original optimization objective is indeed an Integer Linear Program (ILP), which is NP-hard. We explicitly compare our QUBO-based approach with a classical exact ILP solver (via branch-and-bound), and will emphasize this more clearly in the experimental section. The benefit of QUBO is not in replacing linear programming for trivial cases, but in enabling structured quantum-compatible formulations with hardware-aware decomposition, which can become more relevant as hardware capabilities improve.
>
> ### Ablation Studies and Penalty Effects in QUBO:
> We agree that understanding the effect of penalty parameters and hardware constraints is critical. Large penalty weights may increase the dynamic range of the QUBO coefficients, potentially leading to:
> - Weaker embedding quality and increased chain breaks
> - Reduced solution quality due to spectral gap constraints
> - Increased annealing time requirements
>
> These trade-offs are well-known in the quantum annealing literature. While our current contribution focuses on algorithmic structure and theoretical guarantees, we recognize the importance of such ablation studies and plan to pursue them in future work, especially as hardware matures.
>
> We fully agree that quantum annealing is not yet outperforming classical methods in most practical MAPF scenarios. Our paper does not claim otherwise. Instead, our goal is to present a hardware-aware, theoretically grounded framework that integrates quantum optimization in a meaningful way—even if its benefits will become more tangible as hardware improves. We will clarify this positioning and explicitly reference hardware roadmaps such as IBM’s quantum roadmap (https://www.ibm.com/roadmaps/quantum/) to illustrate the forward-looking relevance of this work.
>
> ### Clarification from Eq. (9) to (10):
> Thank you for pointing this out. The equivalence from (9) to (10) stems from the fact that the slack variables in (9) only serve to linearize the inequality constraint Dz ≤ 1. In (10), we exploit the binary nature of D and z, and reformulate the constraint violation as a quadratic penalty without introducing additional variables. We will elaborate on this derivation in the revised version for clarity.
>
> We thank the reviewer again for their helpful suggestions, which we believe will substantially improve the clarity and impact of our paper.

---

### Official Review · Reviewer_uUqH · 2025-03-14

**Overall Recommendation:** 2

**Summary:**

The paper presents a quantum-classical hybrid approach to multi-agent path finding based on solving the restricted master problem via a QUBO translation.

**Claims And Evidence:**

Claims are supported, but the claims are rather weak anyway, involving only certain baseline solvers in a specific setup.

**Essential References Not Discussed:**

none come to mind

**Experimental Designs Or Analyses:**

The comparison study is lacking better baselines as well as a fully classical optimization-based approach. Various choices are insufficiently discussed and the quantum computer setup is not sufficiently explained.

**Methods And Evaluation Criteria:**

The classical baseline approach appears too simple and competitive classical state of the art is not sufficiently discussed.

**Other Comments Or Suggestions:**

typos: "prize" (l. 024), "j--th"(use "$j$th", l. 090), "however" (meant "but", l. 109), wrong citation style (l. 139, e.g.), "implicitely" (l. 227), "c.f." (l. 234), "the in the ..." (l. 359)

**Other Strengths And Weaknesses:**

Aside from what I described above, it is also unclear where any possible advantage up to the point of "dominat[ing] [...] baseline MAPF solvers" should even come from. Quantum annealing is notoriously inefficient at the moment.

**Questions For Authors:**

none

**Relation To Broader Scientific Literature:**

appears fine

**Theoretical Claims:**

One theorem is introduced, although not really needed for the empirical approach of this study.

---

> ### Author Rebuttal · Authors · 2025-03-31
>
> We thank the reviewer for the detailed and structured feedback. We address your concerns below and will incorporate unresolved items into the final version.
> Any other points not mentioned in this answer will be fixed in the camera ready version.
>
> ### On the Strength of Our Claims and Baseline Selection:
> While we appreciate that the reviewer finds our claims supported, we respectfully disagree that they are weak. Our contributions go beyond empirical comparisons:
> - We present the first quantum-compatible optimal MAPF algorithm, introducing a novel integration of QUBO formulations into a branch-and-cut-and-price framework.
> - We provide a theoretical optimality guarantee, generalizing known results to the integer domain.
> - Our hardware-aware QUBO design, incorporating conflict graphs, enables practical decomposition and paves the way for future scalability with emerging quantum hardware.
> - Through our quite general and extensive experimental setup, we show advantages over popular MAPF baselines
>
> ### On Baseline Selection:
> While our experiments compare against selected strong baselines (BCP and LNS2), we agree that more extensive comparisons could further strengthen the work. We are currently running additional experiments with EECBS [1] and LaCAM [2,3], and will incorporate their results or a discussion in the updated version.
>
> ### On Classical Baselines and Fully Classical Solvers:
> The reviewer correctly points out the importance of a fair baseline. In addition to heuristic methods (e.g., LNS2), we note that our ILP-based solver is itself a fully classical method, using branch-and-bound for exact optimization. We will clarify this point in the manuscript and expand the discussion of classical methods in the related work section, including suboptimal but high-performance solvers like EECBS and LaCAM.
>
> ### On Experimental Design and Quantum Annealing Setup:
> We agree that more details on the quantum annealing setup can enhance clarity. We used D-Wave’s default parameters and will include explicit settings such as annealing time (e.g., 50µs) and the number of samples. If the reviewer has specific choices they would like us to elaborate on (e.g., QUBO encoding, annealer scheduling, embedding strategies), we are happy to address those explicitly.
>
> ### On the Usefulness of Quantum Annealing Today:
> We fully agree that quantum annealing is not yet outperforming classical methods in most practical MAPF scenarios. Our paper does not claim otherwise. Instead, our goal is to present a hardware-aware, theoretically grounded framework that integrates quantum optimization in a meaningful way—even if its benefits will become more tangible as hardware improves. We will clarify this positioning and explicitly reference hardware roadmaps such as IBM’s quantum roadmap (https://www.ibm.com/roadmaps/quantum/) to illustrate the forward-looking relevance of this work.
>
> [1] Li, Jiaoyang et al. "EECBS: Bounded-Suboptimal Search for Multi-Agent Path Finding." In Proceedings of the AAAI Conference on Artificial Intelligence (AAAI), 2021.
>
> [2] Okumura, Keisuke. "LaCAM: Search-based Algorithm for Quick Multi-Agent Pathfinding." Proceedings of the AAAI Conference on Artificial Intelligence. Vol. 37. No. 10. 2023.
>
> [3] Okumura, Keisuke. "Engineering LaCAM*: Towards Real-time, Large-scale, and Near-optimal Multi-agent Pathfinding." In Proceedings of the 23rd International Conference on Autonomous Agents and Multiagent Systems (AAMAS), 2024.

---

### Official Review · Reviewer_Hq32 · 2025-03-15

**Overall Recommendation:** 3

**Summary:**

This paper approaches large scale multi-agent path finding as a hybrid quantum computing problem by combining branch-and-cut-and-prize (BCP) with quadratic unconstrained binary optimization (QUBO) formulations for resolving path conflicts. The resulting hybrid algorithms are evaluated on several common large-scale multi-agent path finding tasks, where favorable performance over baseline algorithms is demonstrated.

**Claims And Evidence:**

The claims of the paper are well-motivated and the resulting algorithms are tested on several example scenarios with favorable performance compared to selected baselines. Additional ablations provide further insights into performance differences among e.g. different underlying QUBO formulations.

**Essential References Not Discussed:**

Consider citing a D-Wave technical report.

**Experimental Designs Or Analyses:**

The overall evaluation is good, comparing against key baselines on established benchmarking tasks. Extension to larger scale domains provided in the MAPF benchmark or discussion of why this is currently infeasible would further strengthen the paper.

**Methods And Evaluation Criteria:**

The proposed methods and evaluations make sense, validating performance on established tasks against capable baselines. Scaling to even larger scale examples, as considered by one of the baselines, could further improve the paper (or alternatively further discussion of why this is currently not possible).

**Other Comments Or Suggestions:**

- Line 023: also add CBS abbreviation after “Conflict-based Search”
- Line 131: wait (at) a certain location
- Line 194: NISQ abbreviation used before introduction in Line 306
- The abbreviations QP and QCP might be a bit overloaded for some readers thinking of Quadratic(ally Constrained) Programs - QuP and QuCP could be options?

**Other Strengths And Weaknesses:**

- The paper is well-written, leverages well-crafted illustrations, and is therefore mostly easy to follow
- The maps considered for benchmarking include small to intermediate size scenarios from the MAPF benchmarks by MovingAI. What is restricting extension to the larger maps? Are you hardware limited? It would be interesting to discuss further details here.
- Similarly, the original BCP paper evaluated on the larger den520d and lak503d domains - how would QP-QUBO fare on these tasks?
- The used MAPF benchmark provides different sizes per map type, it would be interesting how performance comparisons behave across map sizes.
- In Figure 3, consider adding environment names to each column in addition to the map
- It could be interesting to see e.g. performance standard deviation around the average in Figure 3, potentially as an extra figure in the appendix to maintain visual clarity.

**Questions For Authors:**

- Could you clarify why LNS2 seemingly improves with increasing agent count on the maze and room domains, while QP-QUBO shows the opposite trend?
- Could you clarify scalability (issues) when considering larger environments from the MAPF benchmark?

**Relation To Broader Scientific Literature:**

The paper gives a good overview of related work and where the current approach is situated relative to prior works.

**Theoretical Claims:**

I did not see issues with the theoretical claims.

---

> ### Author Rebuttal · Authors · 2025-03-31
>
> We thank the reviewer for the thorough and insightful evaluation of our paper. We are encouraged by the positive assessment regarding our motivation, theoretical contributions, experimental validation, and presentation. We address the reviewer’s specific comments and questions below. Everything not addressed in this answer will be integrated into the camera ready version.
>
> ### Scalability to Larger MAPF Instances (e.g., den520d, lak503d):
> We appreciate the suggestion to evaluate on larger-scale maps. The main limitation in scaling to domains such as den520d and lak503d is twofold:
> #### QUBO dimensionality
> The number of binary variables grows rapidly with the number of paths and constraints, which directly impacts the QUBO size and solvability on current NISQ hardware.
> #### Hardware constraints
> Even with conflict graph decomposition, QUBOs derived from larger environments often exceed current quantum annealers' qubit capacities and connectivity constraints.
>
> Our contributions consist of the presentation of the first quantum-compatible optimal algorithm that leverages QUBO
> formulations. Our approach is grounded in a theoretically proven optimality guarantee, which generalizes classical
> column generation theory to the binary domain. We introduce a hardware-aware QUBO design, enabling scalable and
> parallel sub-problem decomposition—crucial for current and near-term quantum devices. By successfully integrating
> these concepts and demonstrating favorable performance on standard MAPF benchmarks, our work paves the way for exploiting future generations of quantum hardware in complex combinatorial planning tasks.
>
> That said, our method is modular and scales well with improvements in hardware or more aggressive decomposition techniques. We plan to explore den520d and lak503d in future work using circuit-based quantum methods or batched hybrid strategies. We will add a discussion on this to the paper.
>
> ### Clarification on QP-QUBO vs. LNS2 Scaling Trends (Maze and Room Domains):
> We agree that the observed trend—LNS2 improving with more agents, while QP-QUBO degrades—is interesting.
> This can be attributed to the heuristic flexibility of LNS2, which benefits from local repair in denser agent settings.
> In contrast, QP-QUBO solves a more rigidly structured optimization problem where QUBO hardness increases with more
> conflicts. Since QUBO is solved heuristically via SA/QA, solution quality can deteriorate with size unless additional
> paths or constraints are added, which also increases problem size.
> Furthemore, we use Prioritized Path Planning for initializing the set of paths which can lead to suboptimal initial
> results for a large number of agents. Adapting the initial heuristic for obtaining a valid set of conflict-free paths (e.g. using LNS2 instead) can
> mitigate this effect. We will clarify these phenomena in the revised version.
>
> ### Benchmark Variability Across Map Sizes:
> Thank you for highlighting this. Due to space constraints, we selected representative maps from each structural class. However, we agree that presenting performance across different sizes within the same map type (e.g., room-32-32 vs. room-64-64) would enrich the analysis. We will consider adding this to the appendix or releasing extended experiments online.

---

> > ### Comment · Reviewer_Hq32 · 2025-04-04
> >
> > Thank you very much for the detailed response and clarifications! Looking forward to the extended "map size" experiments.

---

> > > ### Author Response · Authors · 2025-04-06
> > >
> > > We thank the reviewer for the helpful suggestion for scaling up the experiments. In response, we now include a two-part addition to the revised manuscript:
> > >
> > > ### Empirical Evaluation
> > >
> > > We ran experiments with further state-of-the-art MAPF algorithms EECBS, LaCAM and LaCAM* on benchmark MovingAI maps discussed in the paper.
> > > Similar to the already evaluated BCP and LNS2, we used a maximum time limit of 180s and used default parameters otherwise.
> > > The same setup as in the paper is used: we evaluate each algorithm for 25 different scenarios for 20, 40, 60, 80 and 100 agents.
> > > We compare solving the RMP optimally with ILP (QP-ILP) and with solving the QUBO formulation with simulated annealing (QP-QUBO).
> > > Detailed results indicating mean and standard deviation over 25 different scenarios are given in the tables below for three maps, due to character constraints.
> > >
> > > Considering the mean performance over all 25 scenarios, we observe that our method QP-ILP performs best in 17/20 cases.
> > > It is evident QP-QUBO performs only slightly worse, with also achieving best performance over all baselines in 15/20 cases.
> > > Near future quantum optimization would allow for a performance lying between optimally solving (QP-ILP) and simulated annealing (QP-QUBO).
> > > LaCAM* performs best in 3/20 cases which is probably due to a bad choice of initial paths for our algorithm.
> > > Due to our algorithms' anytime property, it can be combined with other algorithms to also find better initializations (e.g. with LaCAM*).
> > >
> > > Experiments for larger scale maps such as *den520d*, *lak503d* are still running at the moment.
> > > We expect that similar results will be achieved.
> > >
> > > *random-32-32-10*
> > >
> > > | #Agents | LNS2 | EECBS | LaCAM | LaCAM* | QP-ILP | QP-QUBO |
> > > | :-: | :-: | :-: | :-: | :-: | :-: | :-: |
> > > | 20 | 447.1 ± 40.9 | 448.0 ± 40.9 | 453.0 ± 41.0 | 447.1 ± 40.9 | **447.0 ± 40.8** | **447.0 ± 40.8** |
> > > | 40 | 887.3 ± 66.0 | 891.0 ± 65.9 | 910.5 ± 68.2 | 887.1 ± 66.0 | **886.3 ± 66.3** | **886.3 ± 66.3** |
> > > | 60 | 1332.9 ± 77.6 | 1344.9 ± 78.4 | 1388.7 ± 88.5 | 1332.4 ± 77.4 | **1329.6 ± 77.1** | 1330.0 ± 77.0 |
> > > | 80 | 1780.9 ± 95.6 | 1807.4 ± 95.0 | 1880.6 ± 96.3 | 1779.9 ± 94.3 | **1773.6 ± 93.2** | 1775.8 ± 93.9 |
> > > | 100 | 2239.0 ± 113.1 | 2288.5 ± 111.1 | 2410.2 ± 126.5 | 2236.8 ± 112.2 | **2225.4 ± 111.5** | 2231.3 ± 112.2 |
> > >
> > > *room-64-64-8*
> > >
> > > | #Agents | LNS2 | EECBS | LaCAM | LaCAM* | QP-ILP | QP-QUBO |
> > > | :-: | :-: | :-: | :-: | :-: | :-: | :-: |
> > > | 20 | 1208.0 ± 125.9 | 1216.3 ± 127.0 | 1226.6 ± 128.2 | 1203.3 ± 125.2 | **1194.7 ± 124.9** | **1194.7 ± 124.9** |
> > > | 40 | 2455.2 ± 187.5 | 2504.8 ± 189.3 | 2525.9 ± 188.1 | 2434.2 ± 180.0 | **2405.2 ± 178.2** | 2408.2 ± 179.3 |
> > > | 60 | 3673.8 ± 239.5 | 3794.4 ± 258.0 | 3847.4 ± 256.8 | 3634.8 ± 222.6 | **3569.2 ± 216.6** | 3580.2 ± 218.7 |
> > > | 80 | 4992.8 ± 259.1 | 5215.0 ± 307.7 | 5292.2 ± 287.5 | 4927.2 ± 252.3 | **4790.0 ± 969.0** | 4835.7 ± 264.0 |
> > > | 100 | 6301.2 ± 307.2 | 6644.0 ± 342.3 | 6794.6 ± 321.1 | 6219.2 ± 285.4 | **6107.1 ± 303.7** | 6323.4 ± 342.4 |
> > >
> > > *den312d*
> > >
> > > | #Agents | LNS2 | EECBS | LaCAM | LaCAM* | QP-ILP | QP-QUBO |
> > > | :-: | :-: | :-: | :-: | :-: | :-: | :-: |
> > > | 20 | 1059.2 ± 114.0 | 1062.9 ± 115.1 | 1082.0 ± 121.3 | 1059.4 ± 114.1 | **900.5 ± 117.3** | **900.5 ± 117.3** |
> > > | 40 | 2169.0 ± 141.9 | 2183.3 ± 143.3 | 2249.4 ± 160.9 | 2168.8 ± 142.0 | **1815.0 ± 127.5** | **1815.0 ± 127.5** |
> > > | 60 | 3247.2 ± 184.7 | 3287.6 ± 190.8 | 3414.6 ± 208.3 | 3244.6 ± 184.8 | **2727.0 ± 173.7** | **2727.0 ± 173.7** |
> > > | 80 | 4369.3 ± 203.1 | 4444.8 ± 211.3 | 4645.2 ± 252.5 | 4364.2 ± 203.1 | **3627.8 ± 161.2** | **3627.8 ± 161.2** |
> > > | 100 | 5499.4 ± 232.6 | 5641.4 ± 241.1 | 5902.4 ± 273.9 | 5492.3 ± 224.9 | **4558.9 ± 195.2** | 4559.0 ± 195.2 |
> > >
> > > ### Hardware Bottleneck and Forecasting Analysis
> > >
> > > To supplement the above empirical results, we provide a detailed analysis of quantum hardware constraints hindering
> > > evaluating scenarios with thousands of agents and discuss future projections.
> > > - Identification of the main algorithmic bottleneck: Finding a valid solution to the RMP
> > > - Identification of the main hardware bottlenecks: QUBO embedding limitations (qubit count, connectivity) and sample quality (limited by noise and precision).
> > > - A scaling analysis of our problem formulation in terms of qubit requirements (showing how many qubits are needed).
> > > - Reference to realistic near-term hardware improvements, e.g. IBM’s roadmap, and estimates of what scale of MAPF problem such devices might support

---

### Decision · Program_Chairs · 2025-05-01

**Decision:**

Accept (poster)

**Comment:**

The paper presents a hybrid quantum-classical algorithm for solving multiagent path-finding problems. The experiments on quantum hardware show the promise of this approach.

The reviewers acknowledge the work's merits. Their criticisms mostly stem from the paper's originally insufficient explanation of the quantum aspects of this work's background, but the rebuttals have addressed this, and on the whole the reviewers would like to see this paper accepted. The metareviewer encourages the authors to incorporate the explanations they provided during the rebuttal phase into the camera-ready version of the manuscript.